# Exploring unsupervised feature extraction of IMU-based gait data in stroke rehabilitation using a variational autoencoder

Richard Felius[1,2]*, Michiel Punt[1], Marieke Geerars[3], Natasja Wouda[4,5], Rins Rutgers[1], Sjoerd Bruijn[2], Sina David[2], Jaap van Dieën[2]

1 Research Group Lifestyle and Health, Utrecht University of Applied Sciences, Utrecht, The Netherlands,
2 Department of Human Movement Science, Vrije Universiteit Amsterdam, Amsterdam, The Netherlands,
3 Physiotherapy Department Neurology, Rehabilitation Center de Parkgraaf, Utrecht, The Netherlands,
4 Center of Excellence for Rehabilitation Medicine, UMC Utrecht Brain Center, Utrecht, The Netherlands,
5 Department of neurorehabilitation, De Hoogstraat Rehabilitation, Utrecht, The Netherlands

* Richard.Felius@hu.nl

## Abstract

**Data Availability Statement:** All data and software used to process the data and develop the VAE is available on Zenodo (Github) via: https://doi.org/

### Background

Variational AutoEncoders (VAE) might be utilized to extract relevant information from an IMU-based gait measurement by reducing the sensor data to a low-dimensional representation. The present study explored whether VAEs can reduce IMU-based gait data of people after stroke into a few latent features with minimal reconstruction error. Additionally, we evaluated the psychometric properties of the latent features in comparison to gait speed, by assessing 1) their reliability; 2) the difference in scores between people after stroke and healthy controls; and 3) their responsiveness during rehabilitation.

### Methods

We collected test-retest and longitudinal two-minute walk-test data using an IMU from people after stroke in clinical rehabilitation, as well as from a healthy control group. IMU data were segmented into 5-second epochs, which were reduced to 12 latent-feature scores using a VAE. The between-day test-retest reliability of the latent features was assessed using ICC-scores. The differences between the healthy and the stroke group were examined using an independent t-test. Lastly, the responsiveness was determined as the number of individuals who significantly changed during rehabilitation.

### Results

In total, 15,381 epochs from 107 people after stroke and 37 healthy controls were collected. The VAE achieved data reconstruction with minimal errors. Five latent features demonstrated good-to-excellent test-retest reliability. Seven latent features were significantly different between groups. We observed changes during rehabilitation for 21 and 20 individuals in latent-feature scores and gait speed, respectively. However, the direction of the change

10.5281/zenodo.11044903. An interactive version of the VAE is available via: edu.nl/p3kv4.

**Funding:** This study is independent research and was funded by: SIA-RAAK (RAAK.PRO.03.006). SMB was funded by a VIDI grant (016. Vidi.178.014) from the Dutch Organization for Scientific Research (NWO). The funders had no role in study design, data collection and analysis, decision to publish, or preparation of the manuscript.

**Competing interests:** The authors declared no potential conflicts of interest with respect to the research, authorship, and/or publication of this article.

scores of the latent features was ambiguous. Only eleven individuals exhibited changes in both latent-feature scores and gait speed.

## Conclusion

VAEs can be used to effectively reduce IMU-based gait assessment to a concise set of latent features. Some latent features had a high test-retest reliability and differed significantly between healthy controls and people after stroke. Further research is needed to determine their clinical applicability.

## 1. Introduction

One of the primary objectives of stroke rehabilitation is to restore the ability to ambulate in daily life [1]. Gait speed is a frequently used metric to characterize someone's walking ability and to predict if they will be community walkers [2–5]. However, gait speed offers limited insight into the way people walk after a stroke. To gain a more comprehensive understanding of gait recovery, monitor progress and tailor interventions, measuring the way people walk is crucial [6–8]. Inertial Measurement Units (IMUs) are small and portable sensors that enable objective and continuous measurements of gait [9,10]. The main challenge of measuring with IMUs is that the output cannot be interpreted directly, and therefore the data needs to be processed into informative features before IMUs can be effectively employed in research or clinical practice.

Calculating features from IMU data poses several challenges. Firstly, with longer and more complex measurements, e.g., multiple sensors, the number of available features that can be calculated increases. A large number of features makes it challenging to identify the most relevant information and results in a high data redundancy [11,12]. Secondly, features are typically calculated based on theoretical assumptions about what information is most relevant to the study population. However, these assumptions may not necessarily hold true, as there may be useful information in the data that is yet unknown or currently deemed irrelevant.

An alternative approach to obtain features from time-series data that requires fewer theoretical assumptions is the utilization of data-driven methods, which can reduce data to a predefined number of latent features to describe the data. AutoEncoders (AE) are an example of such algorithms [13]. An AE is a model that consists of an encoder, a latent layer with latent features, and a decoder. The encoder reduces the dimensionality of the input data to a set number of latent features in the latent layer. Subsequently, the decoder tries to reconstruct the input data given the latent-feature scores. The AE learns by minimizing the difference between the input and reconstructed data, forcing it to learn a compact, low-dimensional representation of the data. AEs share similarities with principal component analysis (PCA), however, they are capable of modelling non-linear functions as well. The downside of the AE is that it does not constrain the distribution of the latent features, making them unsuitable to generate new data, less robust to input noise, and less reliably with new unseen data. Variational Auto-Encoders (VAE) address the regularization issues of the AE by forcing the latent-feature scores to be normally distributed via an extension of the loss function. In this new loss function, the differences between the distribution of the latent-feature scores and a standard Gaussian distribution are evaluated as well as the differences between the input and reconstructed signal.

Several studies demonstrated the utility of VAEs in analyzing time-series data from electrocardiographic signals and IMUs [14–18]. For instance, Kuznetsov et al. (2021) used a VAE to

encode an electrocardiogram (ECG) into a few interpretable features, each of which represents a distinct aspect of the ECG [15]. Moreover, Fan et al. (2022) applied a VAE to IMU data and improved the accuracy of human activity recognition [18]. It is yet unclear whether a VAE can also be used to obtain relevant information from an IMU-based measurement of the way people walk after a stroke.

The present study aimed to explore if a VAE can be applied to extract a reduced set of latent features from an IMU-based measurement of gait in clinical stroke rehabilitation while maintaining high accuracy in signal reconstruction. Moreover, we aimed to investigate the relevance of the latent features by evaluating the psychometric properties of the latent-feature scores in comparison to gait speed by determining 1) the between-day test-retest reliability of the latent-feature scores; 2) the differences in latent-feature scores between people after stroke and healthy controls; 3) and if the latent features are responsive to changes during rehabilitation.

## 2. Materials and methods

### 2.1 Participants

The study was conducted at five rehabilitation centers in the Netherlands, where data were collected from people after stroke. The dataset included both between-day test-retest measurements and longitudinal data. Additionally, between-day test-retest data were collected from a control group, including adults, and elderly participants at a nursing home. The retest data was measured the subsequent day at approximately the same time of the day. The inclusion criteria for the study were as follows: 1) participants aged 18 years or older; 2) capable of understanding and signing the informed consent document; and 3) able to perform simple tasks. Moreover, people after stroke with first-ever or recurrent stroke were included. Participants were excluded if they were unable to walk at least 0.05 meters per second for two minutes [9]. The medical ethical review committee of Utrecht (METC number: 20-462/C) approved this study. Written informed consent was obtained from all participants involved in the study.

### 2.2 Assessment

Participants walked for two minutes at a self-selected speed on a fourteen-meter walking path with cones at both ends. Data were collected using two unsynchronized Inertial Measurement Units (IMUs) positioned on the left and right foot [19]. The IMUs contained a triaxial accelerometer and gyroscope and measured with a sampling frequency of 104 Hertz (Manufactured by Aemics b.v. Oldenzaal, The Netherlands). The accelerometer and the gyroscope were able to measure up to ±8g, and ±500˚/s, respectively. Participants were allowed to walk with a walking aid. If participants walked with and without walking aid in daily life, the measurement was done twice. These measurements were assumed independent, and both were included in the analysis, since gait is significantly different when walking aided versus unaided [20]. Therefore, both measurements were included. Along with the gait assessment, demographic and stroke-specific characteristics were collected, and participants after stroke underwent several standard clinical tests, including the Berg Balance Scale [21], Trunk Control Test [22], Motricity Index for lower extremities [23], Modified Ranking Scale at admission [24], Barthel Index at admission [25], and the Functional Ambulation Categories (FAC), both with and without a walking aid [26].

### 2.3 Data processing

Following the assessment, the collected data underwent digital processing within an online platform, where they were stored and analyzed. The data processing to compute the gait speed and prepare the data for the VAE is described below.

**2.3.1. Gait speed.** Computation of gait speed was done in seven steps. First, data were downsampled from 104 Hz to 100 Hz. Second, the gyroscope data were corrected for the gyroscope offset derived from a static measurement. Third, a custom-made step-detection algorithm was applied to determine stance and swing phases in the 2MWT (S3 Appendix). Fourth, a sensor-fusion algorithm was used to transform acceleration from a local to a global reference frame by combining the accelerometer and gyroscope data [27]. Fifth, the linear acceleration in the anterior-posterior direct was integrated once to determine the gait speed. Sixth, a Zero-Velocity Update was applied to set velocity to zero during stationary phases of walking, thereby reducing estimation errors [28]. Seventh, the corrected gait speed was integrated to determine position. This calculation enabled us to measure the total distance covered in the two-minute assessment, and thus, the gait speed.

**2.3.2. VAE.** For the VAE, the data processing involved the following steps: first, the data were downsampled from the original recording frequency of 104 Hz to 100 Hz. Second, the gyroscope data were offset corrected using the offset value derived from a static measurement. Third, a step detection algorithm was applied to identify the foot contacts and the stance phases within the gait measurement [9]. Fourth, spanning from the second to second-to-last stride, all 2MWT were split up into epochs of 512 samples, starting in a stance phase, with approximately 50% overlap. Data were split up into segments with 50% overlap to increase the amount of data for the VAE to learn, as a considerable amount of data is required to accurately estimate the model coefficients. An epoch length of 512 samples (5.12 seconds) was chosen to encompass multiple strides per epoch, thus capturing relevant information about an individual's gait pattern. Fifth, a first-order Butterworth bandpass filter with a range of 0.01–10 Hz was applied to filter the data. Sixth, the start of the epochs was set to zero by subtracting the first value. Seventh, for each epoch, the mean and standard deviation were calculated per dimension. Next, the mean and standard deviation of all epochs in the dataset were converted into z-scores. If the mean or standard deviation of a dimension in an epoch had a z-score larger than five, this epoch was deemed an outlier and removed from further analysis. Finally, the epochs were rescaled per dimension with a min-max normalization to a range between -1 and 1, using the minimal and maximal measurable value of the accelerometer [-8g, 8g] and the gyroscope [-500˚/s, 500˚/s].

## 2.4 Model development

A Variational AutoEncoder (VAE) was used to process the IMU data. A VAE comprises two main components: an encoder and a decoder. The encoder maps the input data to a lower-dimensional representation, known as the latent layer, by encoding it into a mean and variance vector. This vector is then used to generate a sample from a probability distribution that models the latent layer. The decoder takes this sample as input and generates a reconstructed output that is similar to the original input data. The difference between the original input and the reconstructed output is measured using a loss function containing the Mean Squared Error and the Kullback-Leibler divergence (KL divergence). A VAE aims to minimize both the Mean Squared Error and the KL divergence, which forces the encoder to learn a good representation of the input data in the latent layer and the latent-feature scores to be normally distributed [13].

The input and output of the VAE used in this study consisted of a 512 X 6 matrix (an epoch), where 512 is the number of data points, and 6 the triaxial acceleration and angular velocity. The encoder and decoder both comprised three mirrored convolutional layers. The latent layer contained 12 latent features. In summary, the VAE learned by reducing an epoch (512X6) to 12 latent features, which are then used to reconstruct the original signal. The VAE

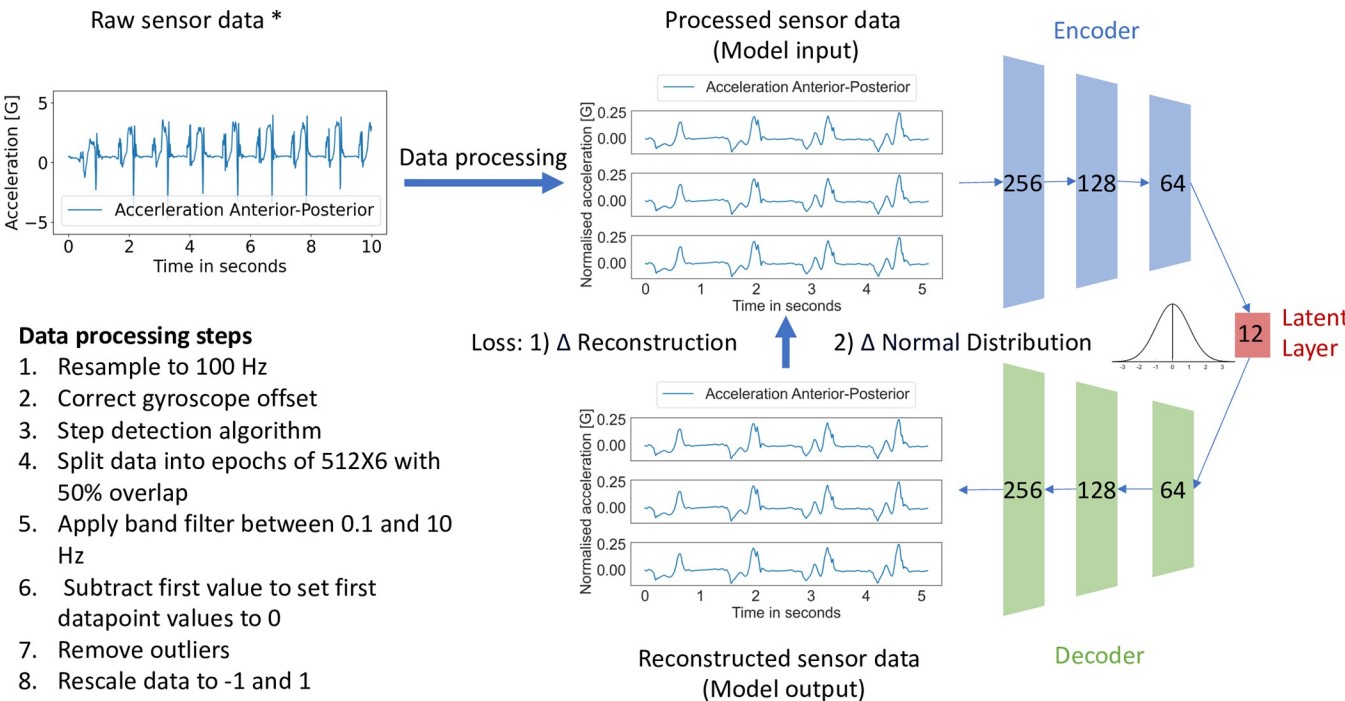

**Fig 1. The hierarchy of the data and the VAE used in this study.** Data were processed and split up into epochs of 512 samples with 6 dimensions (triaxial acceleration and angular velocity). The encoder (green) and the decoder (blue) consisted of 3 mirrored convolutional layers with a size of 256, 128 and 64 nodes. These layers were configured with 32, 64, and 128 filters, respectively, and employed a kernel size of 3. The activation function used throughout the model was a hyperbolic tangent. The latent layer contained 12 normally distributed latent features. The model was trained by comparing the input to the reconstructed output. An Adam optimizer with a learning rate of 0.001 was used. The loss function consisted of two aspects: 1) the difference between the original and reconstructed signal; 2) the difference between the distribution of the latent features and a Gaussian distribution. The VAE was created in Python using TensorFlow version 2.11.0 and is available via: https://zenodo.org/doi/10.5281/zenodo.10878458 [29]. * The actual sensor data consisted of a two-minute measurement with six dimensions. For demonstration purposes a one-dimensional signal was visualized for ten seconds.

tries to reduce the differences between the original and reconstructed signal, while forcing the latent features to be normally distributed. Additional details regarding the chosen number of latent features are provided in S1 Appendix. Fig 1 provides a simplified visual representation of the VAE architecture. The source code of the project is available via: https://zenodo.org/doi/10.5281/zenodo.10878458.

## 2.5 Model evaluation

To avoid any participant-related bias during the training and evaluation process, we used cross-validation [30]. The dataset was split into a training set, a test set, and a validation set on participant level, with a 70/20/10 ratio. This approach ensured that the data from each participant was exclusively used for either training or validation. This process was repeated 10 times so that every participant was included once in the external validation training set. Early stopping was employed if there was no further improvement in performance to prevent overfitting and improve generalization to new, unseen data. The model fit was evaluated with KL divergence, Mean Squared Error and the loss of the external validation data set.

Only the data of the people after stroke were used to train the VAE. As a result, the latent-feature scores represented normally distributed aspects of gait for people after stroke. This allowed us to compare the reconstruction error of the stroke group to the reconstruction error of the healthy group, which might provide some insight into the difference in gait characteristics between healthy controls and people after stroke.

Next, the trained VAE was used to calculate the latent feature scores per epoch. Since one 2MWT measurement consisted of multiple epochs, the average value of the twelve latent features was calculated for both the left and the right foot, resulting in twelve averaged latent feature score per 2MWT. These averaged latent feature scores were used in further statistical analysis.

## 2.6 Statistical analysis

The statistical analysis consisted of three parts. In the first part, the between-day test-retest reliability of the latent feature scores were calculated to indicate if the latent features are consistent. In the second part, the differences in latent feature scores between healthy and individuals after stroke were calculated to determine if the latent features capture information that differs between healthy and stroke. In the third part, the changes in the latent feature scores over time during rehabilitation were determined to indicate if the latent features are responsive to rehabilitation. Fig 2 is a visualization of the type of data that is used to for the creation and evaluation of the model, and to evaluate the reliability, differences and responsiveness.

**2.6.1 Test-retest reliability.**  To assess the test-retest reliability of the averaged latent-feature scores, we utilized the test-retest portion of the dataset. The averaged latent-feature value per measurement of the right foot was used to compute intraclass correlation coefficient (ICC 2.1). An ICC value between 0.5 and 0.75 was considered to indicate moderate reliability, a value between 0.75 and 0.9 was considered to indicate good reliability, and a value greater than or equal to 0.9 was considered to indicate excellent reliability [31]. In addition, we computed

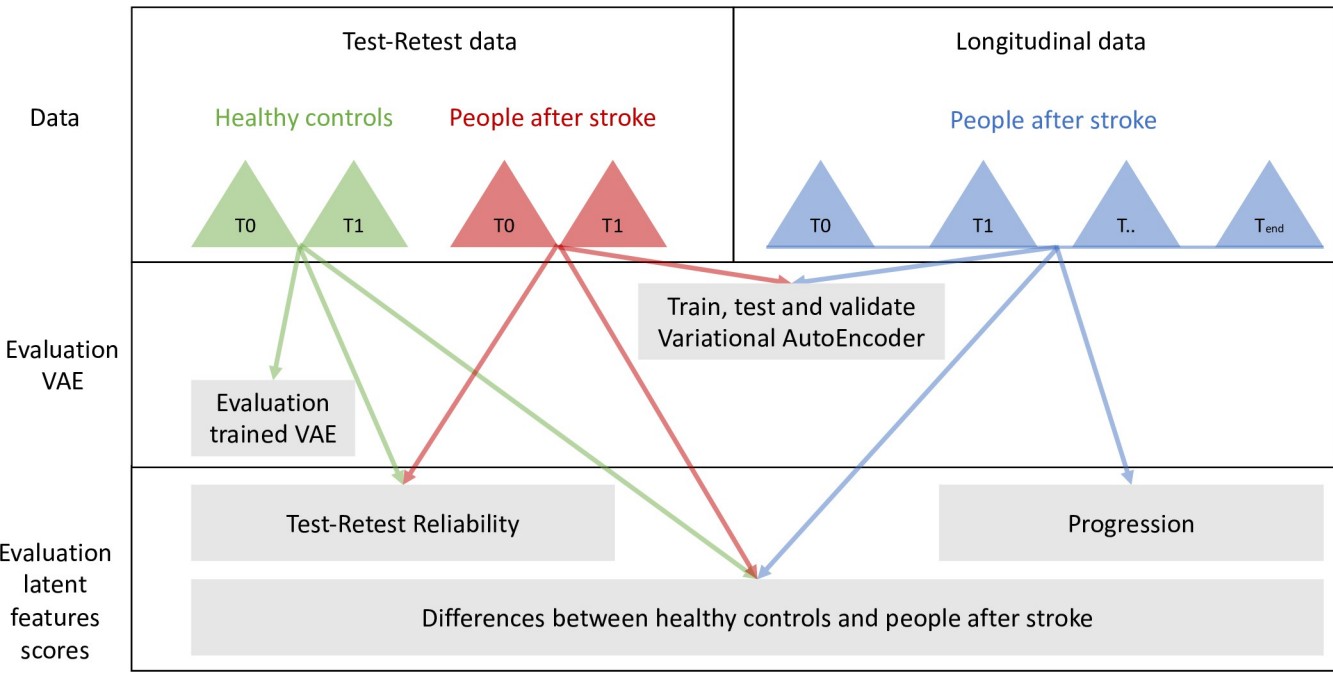

**Fig 2. Visualization of the type of data that is used per research question.** The dataset included both test-retest data from people after stroke (red) and healthy individuals (green), and longitudinal data from people after stroke (blue). The data from the people after stroke was used to train, test, and validate the VAE. The trained VAE was then used to evaluate the model fit, via the reconstruction error, on the data of the healthy control group. Next, the average value per latent feature was calculated for each measurement. These averaged latent features scores were used to 1) determine the between-day test-retest reliability using the test-retest data of the people after stroke and the healthy controls; 2) determine if people after stroke significantly changed during rehabilitation using the longitudinal data; and 3) evaluate the differences between the healthy control group and the stroke group.

the confidence interval (CI), standard error of measurement (SEM), and minimal detectable change (MDC). The MDC represents the magnitude of change in score that exceeds measurement error [32].

**2.6.2. Differences between healthy controls and people after stroke.** The differences between the latent-feature scores and gait speed of the healthy control group and the stroke group were evaluated using an independent t-test [33]. A p-value smaller than 0.05 indicated a significant difference between the healthy and the stroke group. In addition, the effect sizes were calculated using Hedges' g.

**2.6.3. Progression during rehabilitation.** With the MDCs obtained from the test-retest reliability, we determined if participants had significantly changed in a latent-feature score or gait speed during rehabilitation. Only the latent features with good to excellent reliability were included. Moreover, we assessed whether the latent features contained different information regarding recovery than gait speed, by comparing the number of individuals who significantly changed their latent-feature scores to the number of individuals who changed their gait speed.

## 3. Results

### 3.1. Demographics and characteristics

Longitudinal data were collected from 77 people after stroke in clinical rehabilitation. Test-retest data were collected from 30 people after stroke and from 37 healthy individuals. Participant characteristics are described in Table 1. In total, 234 longitudinal and 123 test-retest two-

**Table 1. Characteristics.**

| | | People after stroke (N = 107) | | | Healthy controls (N = 37) | |
| --- | --- | --- | --- | --- | --- | --- |
| | | Longitudinal (N = 77) | Test-retest (N = 30) | | Adults (N = 26) | Elderly (N = 11) |
| Gender | | | | | | |
| | Male | 33 | 15 | | 13 | 0 |
| | Female | 44 | 15 | | 13 | 11 |
| Age | Mean (SD) | 71.5 (12.8) | 69.2 (10.3) | | 42.2 (15.3) | 84.1 (9.0) |
| Walking aid | | | | | | |
| | Yes | 32 | 23 | | 0 | 6 |
| | No | 24 | 4 | | 26 | 5 |
| | Both | 21 | 3 | | 0 | 0 |
| Gait speed [m/s] | Mean (SD) | 0.71 (0.29) | 0.43 (0.24) | | 1.21 (0.13) | 0.83 (0.10) |
| Stroke type | | | | | | |
| | Ischemic | 31 | 24 | | | |
| | Hemorrhagic | 28 | 6 | | | |
| | Unknown | 22 | 0 | | | |
| Stroke side | | | | | | |
| | Left | 31 | 12 | | | |
| | right | 36 | 14 | | | |
| | Unknown | 14 | 4 | | | |
| Barthel index at admission | Mean (SD) | 14.0 (4.6) | 10.3 (4.6) | | | |
| FAC at admission With Walking aid | Mean (SD) | 2.6 (1.8) | 3.7 (0.8) | | | |
| FAC at admission Without Walking aid | Mean (SD) | 2.6 (1.8) | 2.1 (1.6) | | | |

Abbreviations: N = number of participants; SD = standard deviation; m = meters; s = seconds.

minute walk test measurements were included in the analysis. The data processing resulted in 15.505 epochs, with an average of 43 epochs per measurement. After the exclusion of the outliers, 15.381 epochs were included in further analysis, of which 12.747 epochs (82.9%) were data obtained from people after stroke. Thirty-two (10.5%) two-minute walk test measurements from the same individual measured with and without walking aid at one time point.

## 3.2. Evaluation

The VAE was trained and evaluated using data from people after stroke. On average, the training dataset had a size of 8.922 epochs. The test dataset, containing an average of 2.550 epochs, was utilized to assess the model's performance. The validation set, which comprised approximately 1.275 epochs, was used to evaluate the final model fit. This process was repeated ten times so that all people after stroke were once in the external validation set.

The average KL divergence was 0.480 with a standard deviation of 0.157. The average mean squared error was 0.004 with a standard deviation of 0.003. The non-standardized mean absolute error per epoch for the acceleration and angular velocity was 0.15G (±0.05) and 0.59˚/s (±0.18). These results indicate that the model performed well on the external validation set and was able to generalize to new data. In Fig 3, an example of an original and the reconstructed signal is displayed.

**3.2.1 Performance of the variational AutoEncoder.** The trained network was used to determine the reconstruction error of epochs of people after stroke in comparison to a healthy control group. The mean squared error (MSE) per epoch is visualized in Fig 4, in which it is clearly visible that the reconstruction error of the data of the healthy control group is higher than that of the data from people after stroke. This indicates that the VAE is less capable of

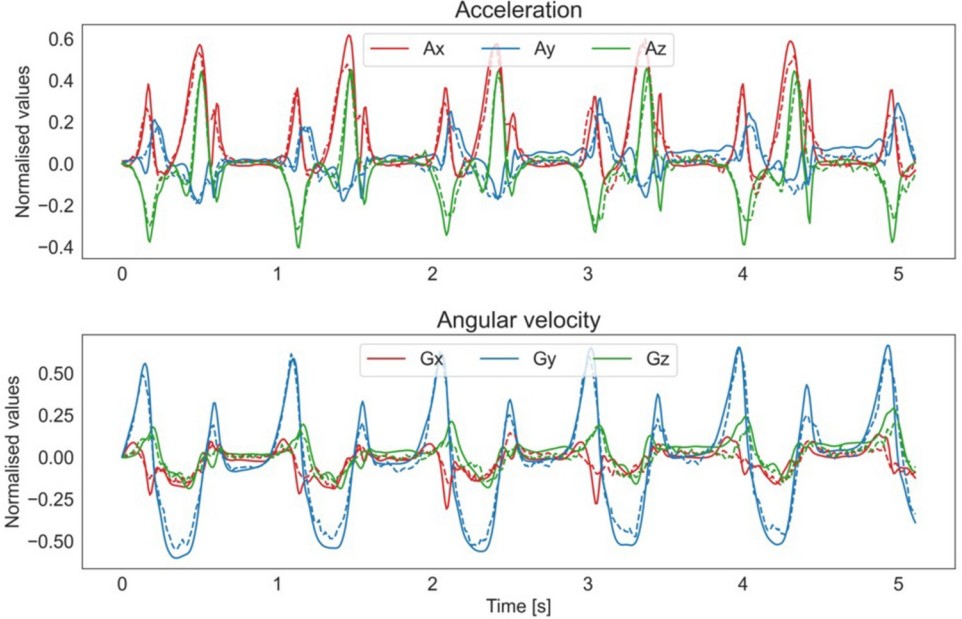

**Fig 3. Example of an original signal and its reconstructed version.** The original signal is a 512 X 6 epoch that consists of a 5.12 seconds (s) measurement with triaxial acceleration (Ax, Ay, Az) and angular velocity (Gx, Gy, Gz) data. The upper panel displays the normalized original and reconstructed acceleration. The lower panel shows the normalized original and reconstructed angular velocity. The reconstructed signal has a strong resemblance to the original signal, as indicated by visual inspection. Overall, this image demonstrates the effectiveness of using a VAE to reduce the dimensionality of a complex signal while maintaining its important features, such as the distinct strides visible in the original signal. More examples are available via: edu.nl/p3kv4.

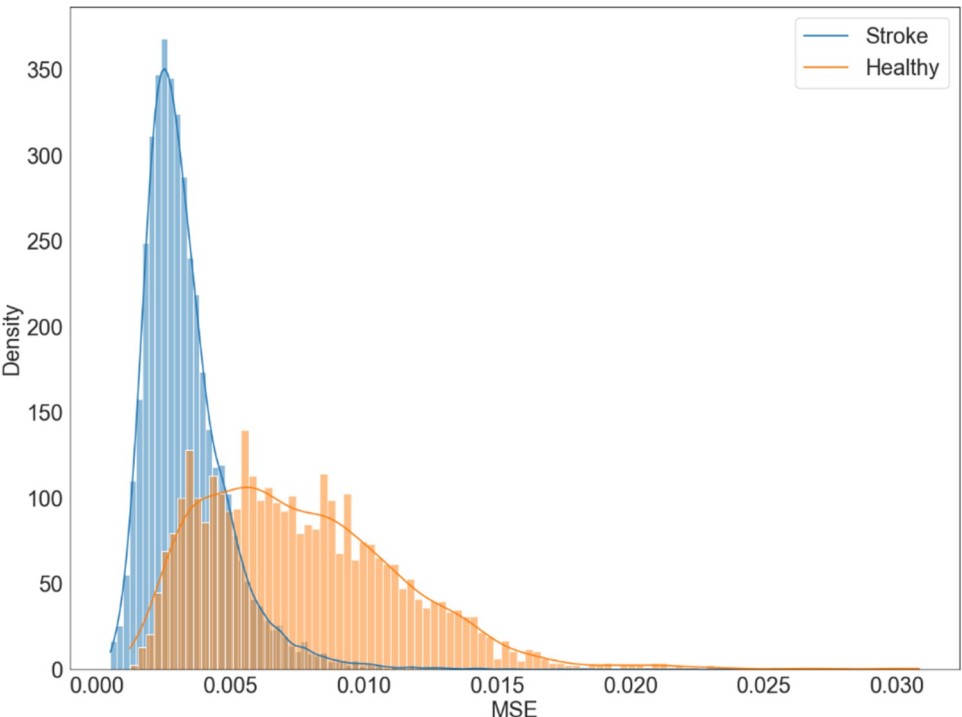

**Fig 4. Distribution of the reconstruction error of gait people after stroke (blue) and healthy controls (orange).**
The reconstruction error was expressed as the Mean Squared error (MSE). The data is normalized on a group level to facilitate comparison between groups. The majority of the epochs from people after stroke were reconstructed with an error below 0.004, while the average reconstruction error of the healthy control group was substantially larger. This indicates that the VAE was less accurate in the data-reduction and reconstruction of the data of the healthy controls.

accurately reducing and reconstructing the data from the healthy control group. Thus, the data from the healthy controls have different characteristics than the data from the stroke group.

**3.2.2. Test-retest reliability.** Twelve test-retest measurements were excluded due to missing data or faulty measurements. Five latent features demonstrated a good to excellent test-retest reliability. The ICC and MDC-values are reported in Table 2. No evident correlation was found between the latent-feature scores and gait speed (S1 Table in S2 Appendix).

**3.2.3. Differences between healthy controls and people after stroke.** Seven latent features indicated a significant difference between the healthy control group and the stroke group. Only four latent features were reliable and significantly different between the groups. Latent feature L1 demonstrated a higher effect size than gait speed. The distribution of all latent features and gait speed is visualized in Fig 5. The P-values are reported in Table 2.

**3.2.4. Progression during rehabilitation.** Admission and discharge measurements of 67 participants were collected. The average outcomes at admission and discharge and the number of participants with increases and decreases on the five reliable latent features are reported in Table 3. On a group level, there were no changes greater than the minimal detectable change. However, some individuals showed significant changes in some latent-feature scores. In total, 30 participants (45%) significantly changed some aspect of their gait. Gait speed appears to be the most responsive, as the highest number of changes were found in this variable. Eleven of the twenty participants who significantly changed their gait speed also changed in the latent features. The other nine participants did significantly change their gait speed. However, this change was not found in the latent-feature scores. Twenty-one participants significantly

**Table 2. Reliability and difference between healthy controls and people after stroke.**

|  | Test-Retest (M = 55) | | Stroke (M = 68) | Healthy (M = 42) | Differences | | |
|---|---|---|---|---|---|---|---|
|  | ICC [CI] | MDC (SEM) | $\mu(\sigma)$[min, max] | $\mu(\sigma)$[min, max] | t-statistic | p-value | Effect size |
| L0* | 0.75 [0.61,0.85] | 2.214 (0.799) | -0.3(0.4)[-1.2, 1.0] | 1.0(1.6)[-1.4, 3.4] | 5.3 | <0.01 | 1.24 |
| L1* | 0.847 [0.75,0.91] | 1.326 (0.478) | 0.1(0.6)[-2.3, 1.2] | -1.7(0.6)[-2.7, 0.0] | -14.2 | <0.01 | 3.00 |
| L2* | 0.851 [0.76,0.91] | 1.126 (0.406) | 0.0(0.6)[-1.9, 1.3] | -0.1(0.8)[-1.7, 1.4] | -0.9 | 0.356 | 0.15 |
| L3 | 0.481 [0.25,0.66] | 1.821 (0.657) | -0.0(0.4)[-0.8, 0.9] | 0.2(0.4)[-0.5, 1.4] | 2.0 | 0.052 | 0.50 |
| L4 | 0.452 [0.21,0.64] | 2.108 (0.761) | -0.0(0.4)[-1.6, 0.6] | 0.4(0.5)[-1.0, 1.3] | 3.6 | <0.01 | 0.91 |
| L5* | 0.899 [0.83,0.94] | 0.831 (0.3) | -0.4(0.4)[-1.3, 0.3] | -1.6(0.9)[-2.9, 0.9] | -7.9 | <0.01 | 1.87 |
| L6 | 0.618 [0.42,0.76] | 1.773 (0.64) | 0.2(0.5)[-1.3, 1.4] | -0.5(0.6)[-1.5, 0.9] | -5.8 | <0.01 | 1.30 |
| L7* | 0.796 [0.67,0.88] | 1.426 (0.514) | -0.2(0.7)[-1.0, 2.1] | -0.5(0.9)[-1.5, 1.6] | -1.8 | 0.072 | 0.38 |
| L8 | 0.684 [0.51,0.8] | 1.197 (0.432) | -0.4(0.6)[-1.7, 0.8] | 0.0(0.3)[-0.5, 0.6] | 5.5 | <0.01 | 0.79 |
| L9 | 0.431 [0.19,0.62] | 2.138 (0.771) | -0.1(0.3)[-1.0, 1.0] | -0.3(0.5)[-1.5, 0.8] | -1.6 | 0.108 | 0.51 |
| L10 | 0.369 [0.11,0.58] | 2.05 (0.74) | 0.1(0.4)[-0.9, 1.4] | 0.2(0.4)[-0.5, 1.4] | 1.5 | 0.132 | 0.25 |
| L11 | 0.704 [0.52,0.82] | 1.208 (0.436) | -0.1(0.4)[-1.0, 0.7] | 0.5(0.5)[-0.4, 1.6] | 6.2 | <0.01 | 1.36 |
| Gait speed* [m/s] | 0.963 [0.92, 0.98] | 0.137 (0.049) | 0.4(0.3)[0.1, 1.1] | 1.1(0.2)[0.6, 1.4] | 15.2 | <0.01 | 2.6 |

Variables with a good to excellent reliability are marked with a *.

Abbreviations: M = measurements; m = meters; s = seconds.

The Test-Retest outcomes of Gait speed has been determined in a previous study.

changed some aspect of their gait, measured with the latent features. Nevertheless, the changes were ambiguous, since some individuals increased, where others decreased.

## 4. Discussion

We explored whether a VAE can be applied to extract a reduced set of latent features from IMU-based measurement of gait in clinical stroke rehabilitation. We found that a VAE can effectively reduce a gait measurement to twelve latent features, as shown by the small differences between the original and reconstructed IMU data. Additionally, we investigated the test-retest reliability and the difference in the latent-feature scores between healthy controls and people after stroke. Four of the twelve latent features were both reliable and were significantly different between groups. Interestingly, one latent feature demonstrated a higher effect size than gait speed in the difference between a healthy control group and individuals after stroke. Furthermore, we evaluated the potential of the latent features to map recovery of gait in addition to gait speed by assessing the changes over time in people after stroke undergoing clinical rehabilitation. We found that approximately 45% of participants significantly changed some aspects of their gait, represented by a change in a latent-feature score or gait speed. However, in contrast to gait speed, the changes in latent-feature scores showed that some individuals increased whereas others decreased over time. This raises the question if the changes in gait, measured with the latent-feature scores, are an indication of between-individual differences in recovery and compensation strategies, i.e. functional adaptations to overcome impairments, or simply due to measurement error [34,35]. For instance, some individuals might reduce gait asymmetry as a result of neuroplasticity, whereas others increase their gait asymmetry with behavioral compensation strategies, both improving function gait. Therefore, people after stroke could change gait in different directions, which causes gait recovery to be a non-linear process.

To date, no study has explored the application of a VAE on IMU-based gait data. However, VAEs have been utilized in other domains involving time-series data [14–18]. For instance,

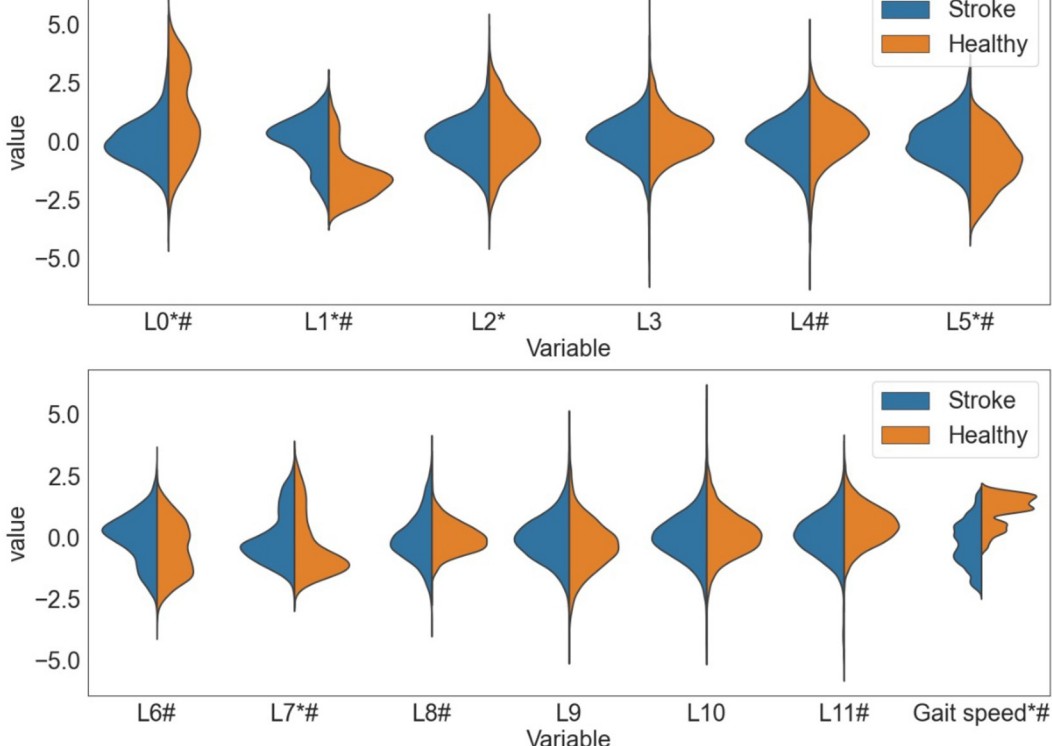

**Fig 5. Distribution of the latent features (L0-L11) and gait speed (z-normalised).** The results of the healthy participants are colored in orange, the results for people after stroke are colored in blue. The * indicates a variable with a high-excellent reliability. The # indicates a significant difference between healthy participants and people after stroke. The height of the distributions on the y-axis indicates the range of the latent variable. The width of the distribution on the x-axis indicates the height of the peak. Since the latent variables are computed with a VAE, the distributions of the stroke group are roughly normally distributed around 0 and are roughly normally distributed. Visual inspection indicates some differences between the healthy and stroke group. First, for the healthy participants, L0 appears to follow a bi-modal distribution. Second, L1 demonstrates a peak at another height than the peak of the stroke group.

Jang et al. (2021) utilized a VAE to detect anomalies in heart rhythm signals, using 60 latent features to reconstruct the signal [14]. Our study demonstrated that fewer latent features are sufficient to achieve accurate gait signal reconstruction, despite dealing with data comprising six dimensions, in contrast to the one-dimensional heart rhythm data. As visualized in S2 Fig in S3 Appendix, the model output improves with more latent features, however, at a decreasing

**Table 3. Changes over time in gait speed and latent-feature scores in people after stroke during clinical rehabilitation.**

| | Longitudinal data | | | | |
|---|---|---|---|---|---|
| | T0 $\mu(\sigma)$[min, max] (N = 67) | Tend $\mu(\sigma)$[min, max] (N = 67) | Increased | Decreased | No change in gait speed |
| L0 | -0.11 (1.05) [-3.19,3.42] | -0.05 (1.06) [-2.85,3.78] | 0 | 0 | 0 |
| L1 | -0.11 (1.05) [-3.19,3.42] | 0.04 (1.11) [-2.97,2.69] | 0 | 3 | 2 |
| L2 | -0.39 (0.95) [-3.12,3.31] | -0.31 (0.99) [-3.38,3.26] | 3 | 3 | 2 |
| L5 | -0.67 (0.66) [-2.98,1.47] | -0.69 (0.71) [-2.98,2.0] | 3 | 3 | 2 |
| L7 | -0.13 (0.99) [-2.55,3.08] | 0.02 (1.12) [-2.38,2.91] | 7 | 3 | 5 |
| Gait speed [m/s] | 0.72 (0.26) [0.07,1.3] | 0.81 (0.26) [0.12,1.36] | 19 | 1 | - |

Abbreviations: N = participants; m = meters; s = seconds.

rate. The downside of utilizing numerous features is that the model is more prone to overfitting. Furthermore, if there are many latent features to evaluate, subsequent analysis requires large datasets, which are often difficult to obtain.

The differences between the original and reconstructed IMU-based gait data were evaluated using the reconstruction error, computed as the mean squared error for the standardized epoch and the average absolute error for both the acceleration and angular velocity. These measures are an indication of the model performance on the whole epoch, thus how well the VAE can reconstruct the data on average. However, in the gait data used, there might be some timeframes in the signal that are more relevant than others. For instance, samples around the foot placement might contain more relevant information that the same number of samples during the stance phase. With the current metrics, it is difficult to determine how well the VAE reconstructs these specific parts of the signal. Moreover, it is unclear if the reconstruction error remains stable over time, thus if the error has the same size during the first hundred samples as the last hundred samples. Future studies should take this into consideration when using a VAE.

The downside of using a deep-learning model, such as the VAE, is that it is difficult to understand which aspects of gait the latent features represent. A more profound understanding of these latent features is crucial for mapping gait recovery and tailoring interventions. One approach to gain a deeper understanding of the information that is captured in a latent-feature score is to adjust the value of a latent-feature score and evaluate the corresponding change in the raw IMU signal. We have developed an online tool, available at: edu.nl/p3kv4, which facilitates this process and enables the user to process a gait epoch, calculate corresponding latent features, and adjust the latent features to evaluate the corresponding changes in the raw signal. This is especially interesting for the latent features that are significantly different between people after stroke and healthy controls, and for the latent features that changed over time during rehabilitation. By using this tool, it is possible to gain some insight into the 'black box' of the VAE and this might increase our understanding of gait recovery in people after stroke.

To evaluate if the latent-feature scores are reliable and can be used to map gait recovery after stroke, the reliability of the latent variable scores was computed with test-retest data of people after stroke in rehabilitation and healthy individuals. The results indicate that only half of the latent-feature scores could be calculated reliably. This might be explained by the fact that some latent-feature scores represent an aspect of the signal that is not necessarily related to the individual walking pattern. For example, in the 2-minute walk test, participants walked straight for 14 meters and then took a turn. Since walking around a cone results in different IMU data than straight walking, it is likely that some features represent turning.

In the current clinical situation, the ability to walk in daily life is assessed using gait speed obtained from a test in a clinical setting, such as a 2-minute walk test. In a previous study (Felius et al., 2023), we evaluated if measuring the way people walk after stroke with IMUs provides information in addition to gait speed [25]. The results of that study suggested that gait speed is strongly associated with most conventional gait features. The present study demonstrated that latent features obtained with a VAE are less correlated with gait speed (S1 Table in S2 Appendix) than conventional gait features. In addition, some latent-feature scores indicated a change over time that was not captured in gait speed. This suggests that the latent features capture different information regarding gait recovery. The difference between the conventional gait features and the latent features might be caused by the inability of the conventional features to fully capture all aspects of the data. Surprisingly, not all individuals who significantly changed their gait speed also showed changes in latent-feature scores. This might be a result of the short length of the included epochs, which was only five seconds, while the gait speed was calculated based on a two-minute measurement.

This study has several limitations. Firstly, some participants completed the two-minute walk test, both with and without walking aids. In our study, we assumed these measurements to be distinct and independent and therefore included both measurements as independent observations in the test-retest and longitudinal data, as walking aids significantly alter an individual's gait [20,36]. However, it is conceivable that these measurements are somewhat correlated. Secondly, we evaluated the reliability and the difference between the healthy and stroke groups using the outcomes of the latent features, while we demonstrated that the reconstruction error of the model on healthy participants was considerably higher than the reconstruction error of data from people after stroke. Consequently, the latent feature scores for healthy controls might lack precision, potentially affecting both the reliability of these scores and the observed differences between healthy and stroke groups. However, if we assume that the reconstruction errors occur randomly, we are more likely to observe smaller differences between groups and lower reliability in our measurements. Thirdly, VAEs are deep learning algorithms that require substantial amounts of data to accurately fit the model. In our study, 12.747 epochs of 107 people after stroke in clinical rehabilitation were used for training, testing, and validation, which may be insufficient to fully leverage the capabilities of the VAE. As a comparison, the study of Jang et al. (2021) had a dataset that was 20 times larger than the dataset we used in this study [14].

There are several opportunities for future studies. First, in the current study, we evaluated the reliability and differences between healthy controls and people after stroke using the individual latent-feature scores derived from the average value per measurement. There might be additional information in the variation of the outcomes of the latent features across all epochs per measurement, which might provide further insights into the variation of gait patterns during a measurement. Moreover, future research should investigate whether the outcomes of the latent features can be utilized to monitor and predict gait recovery or tailored rehabilitation approaches. Lastly, in this study, a VAE was used to extract relevant information from raw IMU-data during gait. Since this study was exploratory, no parameter optimization was applied to achieve optimal model outcomes. Furthermore, there are different types of machine-learning methods, such as t-SNE, that could effectively reduce IMU-data. Future research is required to identify the best type of model and the optimal model architecture to obtain relevant information IMU-based gait data.

## 5. Conclusion

We effectively reduced IMU-based gait assessment to a concise set of latent features utilizing a VAE. Some of these latent features had a high test-retest reliability and differed significantly between healthy individuals and people after stroke. Further research is needed to determine whether and how these latent features can be used clinically.

## Supporting information

**S1 Appendix. Variational AutoEncoder settings evaluation.**
(DOCX)

**S2 Appendix. Correlations.**
(DOCX)

**S3 Appendix. Validity of the stride detection algorithm.**
(DOCX)

## Author Contributions

**Conceptualization:** Richard Felius, Michiel Punt, Marieke Geerars, Natasja Wouda, Sjoerd Bruijn, Sina David, Jaap van Dieën.

**Data curation:** Richard Felius, Marieke Geerars, Natasja Wouda, Rins Rutgers.

**Formal analysis:** Richard Felius.

**Funding acquisition:** Michiel Punt.

**Investigation:** Richard Felius, Michiel Punt, Marieke Geerars, Natasja Wouda, Sjoerd Bruijn, Sina David, Jaap van Dieën.

**Methodology:** Richard Felius, Sjoerd Bruijn, Sina David, Jaap van Dieën.

**Project administration:** Michiel Punt.

**Resources:** Richard Felius, Michiel Punt.

**Software:** Richard Felius, Michiel Punt, Rins Rutgers.

**Supervision:** Michiel Punt, Sjoerd Bruijn, Jaap van Dieën.

**Validation:** Richard Felius.

**Visualization:** Richard Felius.

**Writing – original draft:** Richard Felius.

**Writing – review & editing:** Richard Felius.

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
