## [Decision Letter · Decision Letter 0]

4 Mar 2024

PONE-D-23-38981Exploring Unsupervised Feature Extraction of IMU-Based Gait Data in Stroke Rehabilitation using a Variational AutoEncoderPLOS ONE

Dear Dr. Felius,

Thank you for submitting your manuscript to PLOS ONE. After careful consideration, we feel that it has merit but does not fully meet PLOS ONE’s publication criteria as it currently stands. Therefore, we invite you to submit a revised version of the manuscript that addresses the points raised during the review process.

We look forward to receiving your revised manuscript.

Kind regards,

Jongsang Son, Ph.D.

Academic Editor

PLOS ONE

Journal Requirements:

"This study is independent research and was funded by: SIA-RAAK (RAAK.PRO.03.006). SMB was funded by a VIDI grant (016.Vidi.178.014) from the Dutch Organization for Scientific Research (NWO)."          

5. We note that you have indicated that there are restrictions to data sharing for this study. PLOS only allows data to be available upon request if there are legal or ethical restrictions on sharing data publicly. For more information on unacceptable data access restrictions, please see http://journals.plos.org/plosone/s/data-availability#loc-unacceptable-data-access-restrictions.

Reviewers' comments:

Reviewer's Responses to Questions

**Comments to the Author**

1. Is the manuscript technically sound, and do the data support the conclusions?

Reviewer #1: Partly

Reviewer #2: Yes

2. Has the statistical analysis been performed appropriately and rigorously? 

Reviewer #1: Yes

Reviewer #2: No

3. Have the authors made all data underlying the findings in their manuscript fully available?

Reviewer #1: Yes

Reviewer #2: Yes

4. Is the manuscript presented in an intelligible fashion and written in standard English?

Reviewer #1: Yes

Reviewer #2: Yes

5. Review Comments to the Author

Reviewer #1: COMMENTS TO THE AUTHORS

This study assessed the use of deep learning (VAE) applied to wearable data in the study of stroke patient gait.

Overall, I believe this work has value and I thank the authors for the opportunity to read their work. I have some comments, which I report below, and which I think the authors should consider.

SPECIFIC COMMENTS (line)

TITLE

The title would benefit from including the main finding of the study; for example, can the variational encoder represent movement features well enough?

ABSTRACT

(31-33) Is this the problem addressed? If so, it may help addressing my previous comment about the title. Also, what follows in lines 34-36 is not very clear to me, at least without reading the rest of the manuscript. It looks like you are attempting a classification task here (contrary to the unsupervised approach mentioned earlier and also in the title)? Or you are checking whether there is clustering that could be done and that can discriminate the healthy and pathological group? I suggest rephrasing and making it clearer. Similarly, it would be advisable to better specify what test-retest refers to (different sessions? split in dataset from same session?)

INTRO

(99) The introduction reads well and has a nice flow, leading the reader through the problem under analysis and the broader scope of the work carried out. However, when it comes to the statement of aim and objectives, I suggest more information is provided and more attention is paid to describing the items under investigation. For example, an overview of AE and perhaps most importantly of the distinguishing features of VAE (as opposed to AE) should be provided, as well as a brief discussion about why this architecture has been chosen over others.

Also, more detail would be beneficial re: (1) what type of reliability, (2) what reference for assessing reconstruction capability (is it on original timeseries?), and (3) the nature of the analysis of group (pathological/control) features and monitoring (see comment for Abstract).

METHODS

(110-111) Ditto. Is test-retest from 2 different sessions? How much time apart?

(125) Were 8gs enough to avoid clipping/saturation? Also, what was the rationale behind sensor location? Was it arbitrary, driven by specific reasons (practicality?) or informed by previous work, literature etc.? I appreciate the is no perfect choice and not much knowledge yet about best wearable locations… however it may be beneficial for the reader/other scientist attempting similar approached to know what the thought process was, leading to that selection.

(127) I guess it may not matter much anyway (given the scope of your work and the sample analysed), but I think it may be good to know how many samples of aided/unaided gaits from the same participant were collected (as % of the total). There may be an argument that the features of walking are different, but some characteristic features are carried over for the same individual.

(129) What assessment? Do you mean protocol? In any case, please include a summary here, as readers should not need to access another document to have they key information.

(139) Was 104 Hz imposed by the system used? Does it depend on the no. of units used?

(141-151) Please provide the key aspects of the step-detection algorithm, and its validation outcomes. What is the role of gait event detection in the separation of data? Were epochs 5.12 s long (and therefore included multiple steps/strides?) What is the scope of overlapping and why 50%? Why did not you scale to Z-scores directly? Overall a figure to give a visual description of the steps followed would be beneficial.

(163) Please describe the matrix more clearly. I guess the 6 layers are the 3D accelerations and 3D angular velocities? It is still unclear to me what data goes into the 512 columns, are those all datapoints from the same person? Figure 1 does not help in this sense, and probably should be revised to improve clarity about data input. Similarly, more information should be provided about the selection of the model architecture, and whether any hyperparameter optimisation has been carried out, for example to determine the best configuration of layers, number of nodes per layer, dropout, activation function and, perhaps more importantly, number of latent variables (why 12?). An argument (which I mentioned earlier) would be about the model used… one could ask why VAE? If any other test (other architectures or approaches) has been carried out, it would be good to mention it, so that the journey leading to what presented here is clear (which is informative per se).

(187) Do you mean when there was divergence in performance between training and validation?

(191) If I understand correctly, you are assuming that the differences in the healthy group reconstruction (i.e. real healthy – reconstructed healthy) generated from a model trained with the stroke data only are (or may be) the aspects distinguishing healthy from pathological gait? If so I think:

1) better info should be given since the start of the manuscript (see my previous comments, where I thought you were attempting a classification based on the latent features)

2) some discussion should be presented re: this assumption, as that difference may be the sum of the error from the reconstruction in general (i.e., how good the model is in representing a stroke patient gait) and of the difference between groups? Also, how variance within those two groups is taken into account?

If I have not understood well… then I apologise, but the description may need some change to improve clarity!

(195-200) I do not think this part of the processing is clear enough… and unfortunately Figure 2 does not help much… I suggest rephrasing to improve clarity and revising Figure 2.

RESULTS

(248-249) There is repeated info from methods here.

(251) Are these errors in standardised scores? It would be useful to have an idea of the reconstruction errors in the original units as well… to have a better idea of what they could mean from a biomechanical perspective

(279) The reference to gait speed comes out a bit unexpected, as it was not mentioned earlier in methods.

DISCUSSION

(318) Similarly to what observed for the Results section, it would be better to discuss what is meant for “high accuracy” and its interpretation from a biomechanical perspective (for example, are those differences meaningful when reconstructed as original quantities? What best and worst case scenario, beyond average across gait cycle and multiple people/strides?)

(322) “higher…than speed” between what conditions or groups?

(327-331) These arguments may need some further explanations/examples to make them more accessible to the wider readership

(340) Ok, but is your solution the best compromise? How can you demonstrate it? See previous question about model parameters/architecture and, more broadly the question of whether VAE are better than other dimensionality reduction techniques…

Reviewer #2: Overall, this is an interesting study that provides useful new information to researchers using the IMU technology in the field of stroke rehabilitation. I have a couple of major concerns related to the lack of explanation regarding a statistical analysis that the authors used and justification for 5-second epochs.

Major concerns

The authors did not provide any explanation regarding the statistical analysis that they used in the manuscript. A detailed explanation of their statistical analysis is necessary to be explained in the manuscript.

The data were segmented into 5-second epochs. Why was this segmentation necessary and why was 5-second? The justification for these aspects is necessary to be included in the manuscript.

Minor comments

Lines 99-101: The way you numbered seems confusing. Maybe another way looks better such as 1), 2), 3) or a), b), c).

Line 281: Explanation of abbreviations (L1-8) needs to be added to Table 2 legend.

Some figures missed explanations of abbreviations in their legend. Please double-check and add explanations if needed.

6. PLOS authors have the option to publish the peer review history of their article (what does this mean?). If published, this will include your full peer review and any attached files.

Reviewer #1: **Yes: **Dr Ezio Preatoni

Reviewer #2: No

---

## [Author Response · Author response to Decision Letter 0]

23 Apr 2024

The point-by-point reply is also added as a Word file (formatted version).

Point-by-point reply

We want to thank the editor and the reviewers for their valuable time and insightful feedback. This has helped us to improve the manuscript. In the revised manuscript, we have marked all modifications in red. Below, you will find our responses to each of the comments and questions, organized by number. Each response directly follows the related comment or question. We have also included the sections of text that were copied, altered, and supplemented in response to your feedback below each respective comment or question.

Editor #1

1. Answer:

We have adjusted the manuscript so that it meets the requirements.

2. Answer:

A license has been added to the code, and the code description has been improved. The code is now available on Zenodo with the DOI: 10.5281/zenodo.10878639.

The link to the software and tool has been moved to the data availability statement.

“This study is independent research and was funded by: SIA-RAAK (RAAK.PRO.03.006). SMB was funded by a VIDI grant (016.Vidi.178.014) from the Dutch Organization for Scientific Research (NWO).” 

Please state what role the funders took in the study. If the funders had no role, please state: “The funders had no role in study design, data collection and analysis, decision to publish, or preparation of the manuscript.”

3. Answer:

We have added the following sentence to the manuscript: “The funders had no role in study design, data collection and analysis, decision to publish, or preparation of the manuscript.”

4. Answer:

We have removed the ethics statement from the declarations section.

5. We note that you have indicated that there are restrictions to data sharing for this study. PLOS only allows data to be available upon request if there are legal or ethical restrictions on sharing data publicly. For more information on unacceptable data access restrictions, please see http://journals.plos.org/plosone/s/data-availability#loc-unacceptable-data-access-restrictions.

5. Answer:

We are working on a repository to make the data used in this paper available. We expect to complete this step somewhere in the next two months. Until the de-identified data set is available, the data is available on request via the corresponding author. We have added this information to the data availability section. 

Reviewer #1

1. The title would benefit from including the main finding of the study; for example, can the variational encoder represent movement features well enough?

1. Answer:

Thank you for your suggestion. However, we wanted to emphasize the explorative nature of the study as it is one of the first in this field. Moreover, there is not one main finding or conclusion, since we evaluated both the effectiveness of the dimensionality reduction using a VAE and the psychometric properties of these latent features scores. Therefore, we have decided to keep the title as is.

2. (31-33) Is this the problem addressed? If so, it may help addressing my previous comment about the title. Also, what follows in lines 34-36 is not very clear to me, at least without reading the rest of the manuscript. It looks like you are attempting a classification task here (contrary to the unsupervised approach mentioned earlier and also in the title)? Or you are checking whether there is clustering that could be done and that can discriminate the healthy and pathological group? I suggest rephrasing and making it clearer. Similarly, it would be advisable to better specify what test-retest refers to (different sessions? split in dataset from same session?)

2. Answer:

The main problem addressed here is that there are countless ways to obtain information from complex time-series data, making it difficult to determine which characteristics are the most relevant. Therefore, we explored a different method of extracting relevant information for IMU-data using an unsupervised method and evaluated the psychometric properties of this new method.

We were not attempting to make a classification based on the data. The VAE is used to first reduce the IMU-data to a few latent features, the properties of which are then further explored.

We have rephrased the abstract (see textual changes below) to improve its readability. However, we were limited to the maximum word count, which affected the amount of information that we could share in the abstract. 

2. Textual change (lines 108-122):

“Variational AutoEncoders (VAE) might be utilized to extract relevant information from an IMU-based gait measurement by reducing the sensor data to a low-dimensional representation. The present study explored whether VAEs can reduce IMU-based gait data of people after stroke into a few latent features with minimal reconstruction error. Additionally, we evaluated the psychometric properties of the latent features in comparison to gait speed, by assessing 1) their reliability; 2) the difference in scores between people after stroke and healthy controls; and 3) their responsiveness during rehabilitation.

Methods

We collected test-retest and longitudinal two-minute walk-test data using an IMU from people after stroke in clinical rehabilitation, as well as from a healthy control group. IMU data were segmented into 5-second epochs, which were reduced to 12 latent-feature scores using a VAE. The between-day test-retest reliability of the latent features was assessed using ICC-scores. The differences between the healthy and the stroke group were examined using an independent t-test. Lastly, the responsiveness was determined as the number of individuals who significantly changed during rehabilitation.“

3. (99) The introduction reads well and has a nice flow, leading the reader through the problem under analysis and the broader scope of the work carried out. However, when it comes to the statement of aim and objectives, I suggest more information is provided and more attention is paid to describing the items under investigation. For example, an overview of AE and perhaps most importantly of the distinguishing features of VAE (as opposed to AE) should be provided, as well as a brief discussion about why this architecture has been chosen over others.

Also, more detail would be beneficial re: (1) what type of reliability, (2) what reference for assessing reconstruction capability (is it on original timeseries?), and (3) the nature of the analysis of group (pathological/control) features and monitoring (see comment for Abstract).

3. Answer:

Thank you for your compliment and recommendation. We have extended the introduction to include information about why a Variational Autoencoder was chosen instead of a regular Autoencoder.

Regarding your comment about the model architecture. The goal of this study was to explore the applicability of a deep-learning algorithm on IMU-based gait data, less so to determine the best or optimal model to achieve this goal. 

We chose for a VAE as an unsupervised data-reduction method as it has been used in previous research with time-series data. We think that future studies should try to evaluate different models and model architectures. We have added this as a recommendation in the discussion section.

3. Textual change (lines 223-239; 247-253; 815-820):

“An alternative approach to obtain features from time-series data that requires fewer theoretical assumptions is the utilization of data-driven methods, which can reduce data to a pre-defined number of latent features to describe the data. AutoEncoders (AE) are an example of such algorithms [13]. An AE is a model that consists of an encoder, a latent layer with latent features, and a decoder. The encoder reduces the dimensionality of the input data to a set number of latent features in the latent layer. Subsequently, the decoder tries to reconstruct the input data given the latent-feature scores. The AE learns by minimizing the difference between the input and reconstructed data, forcing it to learn a compact, low-dimensional representation of the data. AEs share similarities with principal component analysis (PCA), however, they are capable of modelling non-linear functions as well. The downside of the AE is that it does not constrain the distribution of the latent features, making them unsuitable to generate new data, less robust to input noise, and less reliably with new unseen data. Variational AutoEncoders (VAE) address the regularization issues of the AE by forcing the latent-feature scores to be normally distributed via an extension of the loss function. In this new loss function, the differences between the distribution of the latent-feature scores and a standard Gaussian distribution are evaluated as well as the differences between the input and reconstructed signal. “

“The present study aimed to explore if a VAE can be applied to extract a reduced set of latent features from an IMU-based measurement of gait in clinical stroke rehabilitation while maintaining high accuracy in signal reconstruction. Moreover, we aimed to investigate the relevance of the latent features by evaluating the psychometric properties of the latent-feature scores in comparison to gait speed by determining 1) the between-day test-retest reliability of the latent-feature scores; 2) the differences in latent-feature scores between people after stroke and healthy controls; 3) and if the latent features are responsive to changes during rehabilitation. “

“Lastly, in this study, a VAE was used to extract relevant information from raw IMU-data during gait. Since this study was exploratory, no parameter optimization was applied to achieve optimal model outcomes. Furthermore, there are different types of machine-learning methods, such as t-SNE, that could effectively reduce IMU-data. Future research is required to identify the best type of model and the optimal model architecture to obtain relevant information IMU-based gait data.”

4. (110-111) Ditto. Is test-retest from 2 different sessions? How much time apart?

4. Answer:

The retest was administered the subsequent day. We have changed the text to between-day test-retest measurements. 

4. Textual change (lines 272-275):

“The dataset included both between-day test-retest measurements and longitudinal data. Additionally, between-day test-retest data were collected from a control group, including adults, and elderly participants at a nursing home. The retest data was measured the subsequent day at approximately the same time of the day.”

5. (125) Were 8gs enough to avoid clipping/saturation? Also, what was the rationale behind sensor location? Was it arbitrary, driven by specific reasons (practicality?) or informed by previous work, literature etc.? I appreciate there is no perfect choice and not much knowledge yet about best wearable locations… however it may be beneficial for the reader/other scientist attempting similar approached to know what the thought process was, leading to that selection.

5. Answer:

Prior to this study, we extensively tested the different sensor settings and locations. We found that 8G and 500�/s were sufficient to avoid clipping while walking in healthy controls and the stroke population. 

Moreover, we chose to place the sensors on the feet, which is a common in research to calculate spatio-temporal parameters, such as gait speed, and because it was the most feasible for our population. We have added a reference to the text.

5. Textual change (lines 284-285):

“Data were collected using two unsynchronized Inertial Measurement Units (IMUs) positioned on the left and right foot [19].”

6. (127) I guess it may not matter much anyway (given the scope of your work and the sample analysed), but I think it may be good to know how many samples of aided/unaided gaits from the same participant were collected (as % of the total). There may be an argument that the features of walking are different, but some characteristic features are carried over for the same individual.

6. Answer:

Around 10% of the measurements were from individuals that walked with and without a walking aid. We have added this information to the results section. Characteristics might indeed be similar within one individual. However, gait with and without a walking aid in an individual after stroke can also be very different as the walking aid offers a considerable amount of stability often resulting in an increase in walking speed. The similarity in features within an individual was also mentioned as a limitation in the discussion.

6. Textual change (lines 538-539):

“Thirty-two (10.5%) two-minute walk test measurements from the same individual measured with and without walking aid at one time point.”

7. (129) What assessment? Do you mean protocol? In any case, please include a summary here, as readers should not need to access another document to have they key information.

7. Answer:

You are correct, it is undesirable to require reading a previous study. We have removed this sentence altogether to avoid confusion, as there is limited extra information available in the other study about the specific protocol.

8. (139) Was 104 Hz imposed by the system used? Does it depend on the no. of units used?

8. Answer:

Yes, the maximum sampling frequency of the used sensor was 104 Hz. Previous work indicated that this frequency was sufficient to reliably measure gait features in people after stroke during rehabilitation. Moreover, we expect the most relevant frequencies to be below 50 Hz (Nyquist Frequency)

9. (141-151) Please provide the key aspects of the step-detection algorithm, and its validation outcomes. What is the role of gait event detection in the separation of data? Were epochs 5.12 s long (and therefore included multiple steps/strides?) What is the scope of overlapping and why 50%? Why did not you scale to Z-scores directly? Overall a figure to give a visual description of the steps followed would be beneficial.

9. Answer:

We have added information regarding the validity of the step detection to the appendix. 

We indeed included sections of 5.12 seconds to include multiple strides in the data, as a shorter duration might be too short to capture relevant information and a longer duration makes it harder for a model to reconstruct the data. We applied a 50% overlap to increase the available data for model training, since a VAE requires a considerable amount of data to train the model.

We used the z-scores on the mean and standard deviation to detect outliers. It is unclear how we would convert the processed data of sample size 512 x 6 to z-scores to perform a different type of outlier detection. 

Lastly, we have adjusted figure 1 to make it clearer how the data was processed, and which data were included in the model.

9. Textual change (lines 320-336; Supplemen

---

## [Decision Letter · Decision Letter 1]

15 May 2024

Exploring Unsupervised Feature Extraction of IMU-Based Gait Data in Stroke Rehabilitation using a Variational AutoEncoder

PONE-D-23-38981R1

Dear Dr. Felius,

We’re pleased to inform you that your manuscript has been judged scientifically suitable for publication and will be formally accepted for publication once it meets all outstanding technical requirements.

Kind regards,

Jongsang Son, Ph.D.

Academic Editor

PLOS ONE

Additional Editor Comments (optional):

Reviewers' comments:

Reviewer's Responses to Questions

**Comments to the Author**

1. If the authors have adequately addressed your comments raised in a previous round of review and you feel that this manuscript is now acceptable for publication, you may indicate that here to bypass the “Comments to the Author” section, enter your conflict of interest statement in the “Confidential to Editor” section, and submit your "Accept" recommendation.

Reviewer #1: All comments have been addressed

Reviewer #2: All comments have been addressed

2. Is the manuscript technically sound, and do the data support the conclusions?

Reviewer #1: Yes

Reviewer #2: Yes

3. Has the statistical analysis been performed appropriately and rigorously? 

Reviewer #1: Yes

Reviewer #2: Yes

4. Have the authors made all data underlying the findings in their manuscript fully available?

Reviewer #1: Yes

Reviewer #2: Yes

5. Is the manuscript presented in an intelligible fashion and written in standard English?

Reviewer #1: Yes

Reviewer #2: Yes

6. Review Comments to the Author

Reviewer #1: Thank you for addressing all my comments.

Nice work, it was a very interesting reading, so I congratulate the team for the research carried out.

Reviewer #2: The authors have addressed all comments appropriately. I don't have further comments. The manuscript is ready to be published.

7. PLOS authors have the option to publish the peer review history of their article (what does this mean?). If published, this will include your full peer review and any attached files.

Reviewer #1: **Yes: **Dr Ezio Preatoni

Reviewer #2: No

---

## [Editor Report · Acceptance letter]

28 May 2024

PONE-D-23-38981R1 

PLOS ONE

Dear Dr. Felius, 

I'm pleased to inform you that your manuscript has been deemed suitable for publication in PLOS ONE. Congratulations! Your manuscript is now being handed over to our production team.

Kind regards, 

on behalf of

Dr. Jongsang Son 

Academic Editor

PLOS ONE